# Synthesis of a Reactive Cationic/Nonionic Waterborne Polyurethane Dye Fixative and Its Application Performance on Viscose Fiber Fabrics

**DOI:** 10.3390/polym16010089

**Published:** 2023-12-27

**Authors:** Changyu Deng, Jiacheng Jin, Hong Zhang, Jiahui Li, Kemei Pei

**Affiliations:** 1School of Chemistry and Chemical Engineering, Zhejiang Sci-Tech University, Hangzhou 310018, China; 202130107284@mails.zstu.edu.cn (C.D.);; 2Zhejiang Litai Composite Material Co., Ltd., Deqing 313000, China

**Keywords:** fixing treatment, cationic waterborne polyurethane, color fastness, viscose fiber fabrics

## Abstract

A series of cationic waterborne polyurethane (CWPU) emulsions was synthesized with isophorone diisocyanate (IPDI) and hexamethylene diisocyanate (HDI) as hard segments; polyol (N210) and polyethylene glycol (PEG-2000) as soft segments; N-methyldiethanolamine (MDEA) as a hydrophilic chain extender; and trimethylolpropane (TMP) as a crosslinker. Then, the effects of the R-value, MDEA content, and TMP content on the properties of the CWPU emulsion, film, and fabric treatment were investigated. The results indicated that when the R-value was 3.0, the MEDA content accounted for 4.0% of the solid and the TMP content accounted for 1.0% of the solid. CWPU has excellent storage stability. Applying it to the fixing treatment of the viscose fiber fabrics can effectively improve the color fastness to rubbing, elasticity, surface smoothness, and anti-static properties.

## 1. Introduction

With the acceleration of globalization, the high quality of textiles has been increasingly valued in the market. Among them, for the quality of textiles, the color fastness of dyed fabrics has been considered to be the most important evaluation index. Viscose fiber fabrics, as significant textile materials, are widely used due to their advantages such as good moisture absorption and breathability, soft handling, and firmness after dyeing. Generally, viscose fiber fabrics can achieve excellent wet rubbing fastness after dyeing with active light-colored dyes. However, achieving the same level of color fastness for dark viscose fiber fabrics proves challenging [1]. The difficulty arises from the presence of a substantial amount of unfixed free dye on the fabric’s surface after dyeing it into a dark color. These free dye molecules contain water-soluble groups like carboxyl and sulfonic groups. Consequently, during the dyeing or washing process, some dyes dissolve in water, resulting in the hydrolysis of the covalent bonds and a weakened bond between the dye molecules and the fabric. This weakened bond makes it easier for the dye molecules to detach from the fabric, leading to a decrease in color fastness [2,3]. Additionally, fabrics with low color fastness can fade after repeated wearing or washing, even contaminating other clothing and causing health issues by transferring dyes to the human body.

To achieve a better fixation effect, it is necessary to utilize dye fixatives to improve the color fastness. The original dye fixatives were mostly formed by the condensation of dicyanodiamide and formaldehyde. However, this dye fixative releases formaldehyde, causing adverse effects on the human body [4]. Currently, formaldehyde-free dye fixatives have received a lot of attention from both academia and industry [5]. However, it is difficult for ordinary formaldehyde-free dye fixatives to improve the wet rubbing fastness to three or more levels. Therefore, the research and development of innovative and efficient dye fixatives holds significant importance [6]. In recent years, waterborne polyurethane has been the focus of color fixing agent research. Its clean and environmental protection characteristics open up a new method for the development of color fixing agents, and it has similar wear resistance, resilience, and adhesion characteristics to those of the traditional solvent-based polyurethane [7,8,9,10,11]. Although waterborne polyurethane exhibits certain color fixing properties and the comprehensive properties of fabrics can be improved by treating fabrics with modified waterborne polyurethane, the current waterborne polyurethane color fixing agents are not enough to replace other types of color fixing agents; so, a lot of research needs to be conducted.

At present, most of the wet rubbing fastness improvers for reactive dyes consist of anionic waterborne polyurethane. However, this type of polyurethane significantly reduces the softness of the fabric because of the interaction between ions and the weak diffusion and adsorption [12]. The structure of cationic waterborne polyurethane (CWPU) contains quaternary ammonium cationic groups, which have the ability to bind with dye anions. This enables the CWPU to form a seamless polymer film on the fabric surface, effectively covering water-soluble dye molecules and preventing detachment. As a result, the color fastness of the fabric is significantly improved [13,14,15,16]. Fan Shaoyu et al. [17] synthesized cationic waterborne polyurethane using isopropanol diisocyanate, hydrophilic chain extender N-methyldiethanolamine (MDEA), and polyethylene glycol as prepolymer monomers. They applied it to the dyeing and fixation of cotton fabrics with reactive dyes, and the color fastness to the friction of the finished fabric reached level 4. Jiang et al. [18] used N-aminoethyl-γ-Aminopropyltrimethoxysilane (KH-792) to modify cationic waterborne polyurethane and were able to improve the friction fastness of dyed chinlon fabric by 0.5 or even 1 level after modification. Chenghao Dong et al. [15], using diisocyanate, polyethylene glycol, butyl ketoxime, and N-methyldihydroxyethyl allyl ammonium chloride (MDAAC) as raw materials, synthesized a new type of environmentally friendly cationic waterborne polyurethane and applied it to cotton fabric. It could increase the wet rubbing fastness of treated cotton fabric from 1~2 to 3.

The specific objectives of this study were as follows. (1) A series of CWPU emulsions was synthesized with isophorone diisocyanate (IPDI) and hexamethylene diisocyanate (HDI) as hard segments; polyol(N210) and polyethylene glycol (PEG-2000) as soft segments; MDEA as a hydrophilic chain extender; and trimethylolpropane (TMP) as a crosslinker. (2) The effects of the R-value (the ratio of isocyanate groups to hydroxyl groups in pre-polymerization), the content of MDEA and TMP, and the addition of MDEA on the properties of CWPU emulsion and film were systematically studied. (3) On this basis, the properties of the emulsion and film were systematically characterized, and the synthetic CWPU dye fixative was used as a fixing treatment for the viscose fiber fabrics; then, the application effect on the fixing treatment of the dark (navy) viscose fiber fabrics was investigated.

## 2. Experiment

### 2.1. Reagents and Instruments

Polyol(N210): technically pure, Jining Baichuan Chemical Co., Ltd., Jining, China; Isophorone diisocyanate (IPDI): technically pure, Shandong Jiaying Chemical Technology Co., Ltd., Shandong, China; Hexamethylene diisocyanate (HDI): technically pure, Guangzhou Disheng chemical Co., Ltd., Guangzhou, China; Dibutyl tin dilaurate (DBTDL): technically pure, Shanghai McLean Biochemical Technology Co., Ltd., Shanghai, China; N-methyldiethanolamine (MDEA): analytical reagent, Shanghai Yien Chemical Technology Co., Ltd., Shanghai, China; Trimethylolpropane (TMP): analytical reagent, Shanghai Yien Chemical Technology Co., Ltd., Shanghai, China; Polyethylene glycol (PEG-2000): Mn = 2000, technically pure, Guangzhou Huahui Chemical Co., Ltd., Guangzhou, China; Butanone oxime: analytical reagent, Shanghai Aladdin Reagent Co., Ltd., Shanghai, China; Ethylenediamine (EDA): analytical reagent, Chengdu Kelong Chemical Co., Ltd., Chengdu, China; Acetic acid: chemically pure, Shanghai Mclean Biochemical Technology Co., Ltd., Shanghai, China; Acetone: analytical reagent, Huzhou Shuanglin Chemical Technology Co., Ltd., Huzhou, China.

Infrared spectrometer: Nicolet iS10, Thermo Nicolet Corporation, Waltham, MA, USA; nanoparticle size/zeta potential analyzer: SZ-100V2, Horiba Trading Co., Ltd., Shanghai, China; digital viscometer: NDJ-9S, China Bangxi Instrument Technology Co., Ltd., Shanghai, China; color fastness to rubbing tester: Y(B)571-Ⅲ, China Darong Textile Instrument Co., Ltd., Wenzhou, China; thermogravimetric analyzer: TGA550, TA Instruments Waters Corporation, Shanghai, China; fabric crease recovery tester: YG541E, China Jigao Testing Instruments Co., Ltd., Wenzhou, China; Haier drum washing machine: XQG 80-BX 12636, China Haier Zhijia Co., Ltd., Qingdao, China; Martindale wear and pilling tester: TF10, Tesda Corporation, Zurich, Switzerland; fabric colorimeter: X-Rite Color i7, Aiselei Company, USA, Ann Arbor, MI, USA; textile fabric test instrument: YG541E, China Depu Textile Technology Co., Ltd., Changzhou, China.

### 2.2. Experimental Procedure

Prior to the experiment, the N210 underwent dehydration at a temperature of 120 °C and a pressure above 0.1 Mpa for a duration of 6 h. Afterwards, it was cooled to room temperature. The MDEA and PEG-2000 were dried in a constant-temperature oven at 120 °C for 1 h. For the experiment, pre-weighed amounts of N210, HDI, and IPDI were added to a dry four-mouth flask fitted with a temperature-controlled, stirred, and reflux condensation device. The mixture was stirred at room temperature for 0.5 h and then heated to 110 °C. DBTDL was added drop by drop with mechanical stirring for a duration of 3 h. In the second step, the temperature was lowered to 60 °C. The reaction was conducted for 5 h after adding the MDEA solution, TMP, and PEG2000 dropwise. Subsequently, butanone oxime was added, followed by the addition of EDA; acetic acid and deionized water were added and dispersed at high speed. By removing acetone from the system through vacuum distillation, the reactive cationic aqueous polyurethane fixing agent CWPU could be obtained. The synthesis route is shown in Figure 1.

### 2.3. Preparation of CWPU Film

The process involved utilizing a clean and dry polytetrafluoroethylene template to pour a waterborne polyurethane emulsion, creating a consistent and thin layer. The excess material was carefully removed using a glass rod. Following this, the thin layer was left to settle at room temperature for a period of 24 h before being dried at 160 °C for a duration of 8 min. As a result, a CWPU film was successfully formed, measuring approximately 0.3 mm in thickness.

### 2.4. Technological Process

First, the viscose fiber fabric was immersed in distilled water for 5 min. Following this, it was then immersed in CWPU dye fixative, which had a concentration of 20 g/L, along with an HDI dilute solution containing mass fractions of 0%, 3%, and 5%, respectively. This process was carried out at room temperature for 0.5 h, with the purpose of eliminating any excess moisture (residual rate of 80%).

### 2.5. Characterization

FTIR. The Nicolet iS10 (Thermo Nicolet Corporation, Waltham, MA, USA;) infrared spectrometer was used for testing; it performed infrared spectroscopy testing on the film in ATR mode with a wavenumber collection range of 400–4000 cm^−1^.

Thermal Characterization. The TGA550 thermogravimetric analyzer (TA Instruments Waters Corporation, Shanghai, China;) was used for testing; the thermogravimetric analysis was completed under N2 protection, with a temperature range of 30–600 °C and a heating rate of 10 °C/min.

Emulsion Particle Size. Tests were carried out using the nanoparticle/zeta potential analyzer SZ-100V2 (Horiba Trading Co., Ltd., Shanghai, China). The particle size test method was used for performance testing. We diluted CWPU into a 1% solution with deionized water and measured the particle size using a nanoparticle size zeta potentiometer (Horiba Trading Co., Ltd., Shanghai, China) and repeated the process three times to obtain an average result.

Viscosity. The digital display viscometer NDJ-9S (China Bangxi Instrument Technology Co., Ltd., Shanghai, China) was used for testing; a 50 mL sample was placed in a beaker at room temperature for viscosity testing; the beaker was rotated for 1 min; the date was read; each sample was measured 3 times, and the average value was calculated. Referring to the standard GB/T 2794-2022 [19] a certain amount of sample was taken and placed in a glass sample bottle with the lid tightly closed. It was stored at 60 °C for 7 days and observed for sedimentation, layering, and other phenomena.

Water absorption. We prepared 2 cm × 2 cm CWPU film, weighed it, and recorded the weight of the sample before water absorption. Next, we submerged the film in deionized water and sealed it at room temperature for a duration of 24 h. After the specified time, we removed the film from the deionized water, dried it quickly using filter paper, and weighed it again. The water absorption rate of the film was calculated according to the following formula:(1)$$q=(m_{n}-m_{0})/m_{0}\times 100\%$$
where q is the water absorption rate (%); m_n_ is the sample mass after water absorption (g); and m_0_ is sample mass before water absorption (g).

Crease recovery test. The fabric crease recovery tester YG541E (China Jigao Testing Instruments Co., Ltd., Wenzhou, China) was used for testing; the crease recovery was tested according to the “Textile fabrics-Determination of the recovery from creasing of a folded specimen by measuring the angle of recovery” (GB/T 3819-1997 [20]).

Rubbing fastness tests. The friction fastness tester Y(B)571-III (China Darong Textile Instrument Co., Ltd., Wenzhou, China) was used for testing; the color fastness to rubbing was tested according to the “Textiles-Tests for color fastness-Color fastness to rubbing” (GB/T 3920-2008 [21]) and evaluated according to the “Textiles-Tests for color fastness-Grey scale for assessing staining” (GB/T 251-2008 [22,23]).

Appearance flatness test. The textile fabric testing instrument YG541E (China Depu Textile Technology Co., Ltd., Changzhou, China) was used for testing; the smoothness appearance referred to the “Textile-Test method for assessing the smoothness appearance of fabrics after cleaning” (GB/T 13769-2009 [24]).

Test for resistance to fuzz and pilling. The Martindale wear and pilling tester TF210 (Tesda Corporation, Switzerland) was used for testing; the anti-fuzzing and pilling performance was tested according to the “Textiles-Determination of fabric propensity to surface fuzzing and to pilling-Part 2: Modified Martindale method” (GB/T 4802.2-2008 [25])

Color difference test. The Haier drum washing machine XQG 80-BX 12636 (China Haier Zhijia Co., Ltd., Qingdao, China) was used for testing; the specific parameters of the fabric washing program were the washing temperature of 30 °C, three washing cycles, and drum drying (68 °C). X-Rite Color i7 (Aiselei Company, USA, Ann Arbor, MI, USA) was used to test the color change index and the yellowing index of the fabrics.

## 3. Results and Discussion

### 3.1. Optimization of Synthesis Process

#### 3.1.1. The Effect of R-Value on the Performance of CWPU

The R-value is the molar ratio of -NCO and -OH in the prepolymer. Generally, the molecular weight of the prepolymer and the ratio of the soft and hard segments can be adjusted by changing the R-value and thereby changing the performance of the CWPU. In this experiment, the MDEA content was 4.0% and the TMP content was 0.0%. A series of CWPU emulsions was synthesized by only changing the R-value. Its influence on the performance of the CWPU emulsion is shown in Table 1 and Table 2 and Figure 2.

Table 1 and Table 2 shows that with the increase in the R-value, the viscosity gradually decreased from 387 mPa·s to 264 mPa·s. The appearance of the emulsion changed from translucent to transparent pan-blue light. The storage stability gradually improved until the R-value exceeded 3.5. The decrease in the viscosity of the emulsion was attributed to the increase in the R-value. As the R-value increased, the molecular weight of the prepolymer decreased and the molecular chain became shorter. Consequently, the interaction between the molecular chains was weakened, leading to a reduction in internal friction resistance between particles. This allowed the emulsion to be more evenly dispersed in water during the emulsification process, and the particles no longer aggregated, thus reducing the viscosity of the emulsion [26]. In the event that the R-value is insufficient, the prepolymer’s molecular weight will be excessive, leading to elongation of the molecular chain. Consequently, this will result in elevated viscosity and challenges in achieving emulsification and dispersion, as well as compromised stability. Conversely, if the R-value was excessive, the residual isocyanate reacted with water to produce amines, causing the formation of insoluble polysubstituted ureas that were prone to delamination [27]. It can be seen from Table 1 and Table 2 and Figure 2 that the water absorption of the film exhibited a decreasing trend initially as the R-value increased. However, when it reached a certain point, the water absorption started to increase again. Additionally, the texture of the film gradually transitioned to a softer and less viscous state. The increase in the R-value resulted in a higher proportion of hard segment content within the prepolymer. This increase enhanced the interaction force between the molecular chains, leading to a relatively dense film formation. Consequently, it became challenging for the water molecules to permeate the film, resulting in a gradual decrease in the water absorption rate and an improvement in water resistance. However, an excessively large R-value led to shorter molecular chains, which hindered the production of dense films. Moreover, it facilitated water molecule penetration into the film, causing an increase in the water absorption rate and a decrease in water resistance.

#### 3.1.2. The Addition Method of Hydrophilic Chain Extender MDEA

MDEA is a quaternary ammonium compound. It is not only a hydrophilic chain extender, but also a catalyst for the reaction of -NCO with active hydrogen compounds. Adding MDEA to the system will accelerate the reaction between the isocyanate groups and hydroxyl groups. But if the control is improper and the reaction speed is too fast, the reaction system will easily lead to gel. In this part of the optimization experiment, the R-value was adjusted to 3.0, the content of MDEA was set to 4.0% and the content of TMP was set to 1.0%. The focus was solely on modifying the method of MDEA addition and studying its impact on the emulsion state. The results are shown in Table 3.

When MDEA was directly added, the hydroxyl content increased sharply and reacted rapidly with isocyanate. The response generated considerable intensity, resulting in rapid growth of the chain segment and a significant increase in viscosity. This heightened reactivity poses a risk of gel formation, as shown in Table 3. In the MDEA chain expansion stage, the most suitable addition method was to add dropwise for 20 to 30 min. The increase in the hydroxyl group content was gradual due to the slow drop rate, while the reaction speed with isocyanates was rapid. MDEA fully reacted in an instant, and the concentration of MDEA in the system tended to zero. The materials were like a “hungry state”. The heat released from the reaction was easy to control, and it could also disperse uniformly after emulsification [28]. Moreover, the addition of acetone to MDEA effectively decreased the emulsion’s viscosity, resulting in a decreased reaction rate and preventing implosion. In summary, the preferred addition method for this experiment was to dissolve MDEA in acetone and add dropwise for 20 to 30 min.

#### 3.1.3. The Effect of MDEA Content on the Performance of CWPU

Hydrophilic chain extenders can introduce hydrophilic groups into the system, which is conducive to more uniform dispersion in water during emulsification. Introducing the hydrophilic chain extender MDEA into the polyurethane molecular chain can result in functional monomers on the polyurethane chain segments that can be ionized. In this part of the experiment, the R-value was set to 3.0, the TMP content was adjusted to 1.0%, and the MDEA was added dropwise for 20 min. The performance of the CWPU emulsion was evaluated by analyzing the impact of the variations in the MDEA content. The findings are presented in Table 4 and Table 5 and Figure 3.

According to the data presented in Table 4 and Table 5, the viscosity of the emulsion showed a pattern of initially decreasing and then increasing as the MDEA content increased. The reason was that when the content of MDEA was small, the hydrophilic groups in the system were small, and the particle size was large, which made the viscosity of the emulsion smaller. As the MDEA content increased, the proportion of quaternary ammonium cations in the dispersion system also increased, leading to an increase in particle volume and thus viscosity [29]. If the content of MDEA was too small, the emulsion was not stably dispersed in water. With the increase in MDEA content, the appearance of the emulsion changed from semitransparent to transparent pan-blue light and then became transparent. The water solubility of the waterborne polyurethane increased with the higher MDEA content, resulting in a more transparent and stable emulsion [30]. On the one hand, the increase in hydrophilic groups increased the hydrophilicity of the molecular chain segments, making it easier for them to uniformly disperse in water during emulsification. On the other hand, due to the positive charge in the molecular chain, the electrostatic force between each segment also increased. The phase inversion process was facilitated during emulsification, leading to enhanced stability [31]. From Table 4 and Table 5 and Figure 3, it can be concluded that as the concentration of MDEA increased, there was a gradual increase in the water absorption of the film, and the feel of the film changed from soft and non-sticky to sticky. Throughout the immersion of the film in water, there was an observable augmentation in the hydrophilic groups. This resulted in a heightened possibility for water molecules to bond with hydrophilic ions within the molecular chain, consequently leading to an escalation in the rate of water absorption [32].

#### 3.1.4. The Effect of TMP Content on the Performance of CWPU

Linear waterborne polyurethane has many defects, such as poor mechanical properties. The appropriate amount of crosslinker had been introduced and can synthesize CWPU with a network structure, which can effectively improve the performance of waterborne polyurethane film [33]. In this part of the experiment, the R-value was set to 3.0, and the MDEA content was adjusted to 4.0%. Using the addition method, the MDEA was added dropwise for 20 min, but this only changed the TMP content. The influence of TMP content on the performance of CWPU emulsion is shown in Table 6 and Table 7 and Figure 3.

It can be concluded from Table 6 and Table 7 that with the increase in TMP content, the appearance of the emulsion was a transparent pan-blue light and the viscosity increased gradually from 290 mPa·s to 345 mPa·s. The reason was that with the addition of TMP, the degree of crosslinking of the polymer was deepened, the chain segments changed from a linear structure to a network structure, and the molecular weight became larger, making it more difficult to emulsify in water, which led to an increase in the internal friction resistance between the molecular chains and a restriction in the movement, which thereby led to the increase in the viscosity of the emulsion. The instability of the emulsion was observed when the TMP content reached 4.0%. This can be attributed to the high molecular weight of the polymer and the resulting increased crosslinking density. Consequently, the dispersion of the emulsion with water became challenging, leading to a decrease in stability. It can be seen from Table 6 and Table 7 and Figure 4 that when the TMP content was 1% to 2%, the water absorption rate of the film was smaller than that of the WPU without TMP, and the water resistance of the film was enhanced. When the content of TMP exceeded 2%, the film’s water absorption increased, and its water resistance was compromised. The increase in TMP content led to the formation of a crosslinked structure in the polyurethane molecule. This, in turn, enhanced the crosslinking density of the polyurethane chain. As a result of this crosslinked structure, the membrane became denser and exhibited lower permeability to water molecules. Consequently, there was a decrease in water absorption. However, when the TMP content was increased to more than 2%, the crosslinking density was too large, and the polymer chain could not be uniformly carried, resulting in greater water absorption. The feel of the film gradually became soft and non-sticky. The main reason was that with the introduction of TMP, the crosslinking degree of the polymer increased, forming a network structure, which made the film less sticky.

### 3.2. Characterization of CWPU Emulsion and Film

#### 3.2.1. FT-IR

Infrared spectrum testing was conducted on the optimal synthesized CWPU dye fixative film, and the results are shown in Figure 5.

There was no obvious -NCO characteristic peak at 2243 cm^−1^, indicating that all the -NCO groups had reacted. There was no obvious absorption band at around 3590 cm^−1^, indicating that the hydroxyl groups in the system were basically involved in the reaction. The peaks at 2937 cm^−1^ and 2852 cm^−1^ were the stretching vibration peaks of -CH_2_ and -CH_3_. The peak at 1531 cm^−1^ was generated by a N-H vibration; the peak at 1715 cm^−1^ was a C=O absorption peak, and the peak at 1238 cm^−1^ was an amide C-N stretching vibration peak. These indicate that the synthesized polymer had a -NHCOO- structure. Furthermore, the presence of a prominent peak at 1103 cm^−1^, corresponding to C-O-C, confirmed that the polyurethane produced was a waterborne polyether polyurethane. This explanation served as evidence of the successful creation of the CWPU film.

#### 3.2.2. TG

The thermal stability of polyurethane can be measured by its thermal decomposition temperature. Thermogravimetric analysis spectra generally include two kinds of curves: one is the curve of the sample mass retention fraction and temperature (TG curve); the other is the curve of the rate and temperature of the sample weight change (DTG curve). The thermal stability of polyurethane largely depends on the structure of the soft segment. The hard segment is basically a fixed phase, which is relatively regular, with a high intercrystalline force and good thermal stability. The thermogravimetric analyzer (TA Instruments Waters Corporation, Shanghai, China) was used to test the optimized synthetic CWPU film, and the analysis results are shown in Figure 6.

The thermogravimetric (TG-DTG) curve of CWPU is shown in Figure 6. The thermal decomposition temperature of cationic waterborne polyurethane is about 270 °C, and it has good heat resistance. The quality loss rate was low when the temperature was lower than 270 °C. This might be due to the evaporation of water from the membrane and the thermal decomposition of a small number of small molecule impurities. At temperatures ranging from 270 to 360 °C, there was a significant decrease in the polymer’s mass. This phenomenon could be attributed to the initiation of molecular movement at the chain end, the gradual untangling of the polymer chains, and the onset of decomposition in the shorter chain molecules. The DTG curve exhibited a significant exothermic peak within the temperature range of 260~400 °C. During this stage, the polymer initiated a rapid decrease in weight, while the rigid segment of the polymer chain simultaneously underwent decomposition. Consequently, this resulted in the opening of the main chain and the transformation of the elongated chain’s flexible section into a shortened chain structure due to thermal degradation. When the temperature reached a level above 360 °C, the polymer was completely decomposed into small molecular compounds. The degradation temperature of the polymer could be used to measure the heat resistance of the polymer. The initial degradation temperature of the CWPU film synthesized in the experiment was 270 °C, indicating that the heat resistance of the film was good.

#### 3.2.3. Particle Size of Emulsion

We diluted a quantity of waterborne polyurethane dye fixative emulsion with deionized water at a ratio of 1:100. Subsequently, we analyzed the emulsion’s particle size using a nanometer analyzer (nanoparticle size/zeta potential analyzer: SZ-100V2, Horiba Trading Co., Ltd., Shanghai, China). The results are shown in Figure 7. The particle size has an important influence on the properties of waterborne polyurethane emulsion and film. On the one hand, as a heterogeneous dispersion system, the appearance, stability, and other properties are closely related to particle size and distribution. On the other hand, the particle size of the emulsion and the distribution determine the stacking mode between the emulsion particles, the degree of molecular chain diffusion, and the uniformity of the film structure during film formation. Therefore, the film properties, such as water resistance, and the mechanical properties are closely related to the particle size and the distribution of waterborne polyurethane [34]. From Figure 7, it is evident that the average size of the CWPU particles was measured to be 34 nm. These particles were primarily distributed within the range of 35–50 nm, displaying a concentrated peak distribution. This observation suggests that the synthesized CWPU possessed a consistent and reliable size distribution. The reason behind this uniformity can be attributed to the incorporation of MDEA, which enhanced the hydrophilicity of the molecular chains. As a result, the hydration of CWPU was promoted, leading to a reduction in molecular chain entanglement and facilitating the microphase separation of the polymers. On the other hand, the introduction of cationic groups led to an increase in the charge density on the surface of the emulsion and a decrease in particle size.

### 3.3. Evaluation of Application Performance

The “two soaking and two pricking” process was used to fix the treatment of the viscose fiber fabrics, and the effect of the HDI mass fraction on the friction color fastness and K/S value of the fabric treated with waterborne polyurethane dye fixative was studied. The dosage of waterborne polyurethane dye fixative was 20 g/L. The results are shown in Table 8.

According to the data presented in Table 8, it is evident that the color fastness to dry friction of the viscose fiber fabrics could reach levels of 4–5, while the color fastness to wet friction could reach level 3. The change in K/S value was not obvious. The chromatism of PUD-2 and PUD-3 after fabric finishing was less than 0.5, while the PUD-1 was relatively large, reaching 0.63. Compared with the original fabric, the color fastness to the friction of the fabric after the fixation treatment increased. The cationic waterborne polyurethane contains a quaternary ammonium group which had the ability to establish an electrostatic binding with the anions present on the fiber. This interaction resulted in the formation of insoluble color lakes, enabling the dye and dye fixative to effectively deposit onto the fiber [35]. The anti-pilling and anti-fuzzing performance of PUD-1, PUD-2, and PUD-3 was level 4. Compared with the original fabric, the performance after the fabric was treated with three samples was significantly improved. This was because the introduced CWPU forms a dense adhesive film to protect the fibers, preventing external damage to the film and improving the fabric’s anti-pilling and anti-fuzzing performance. The smoothness of PUD-1, PUD-2, and PUD-3 was enhanced, resulting in an improved appearance compared to the original fabric. This was due to the mutual exclusion and thermodynamic incompatibility between the soft and hard segments of the CWPU, which exhibited different morphological characteristics with temperature changes and thus had a certain impact on the conformality of the fabric [36]. The front rebound angles of the fabric in the meridional and latitudinal directions are shown in Table 8. Here, the total crease recovery angle was the sum of the meridional crease recovery angle and the latitudinal crease recovery angle. The results demonstrate that the crease recovery angle of PUD-1, PUD-2, and PUD-3 was greater than that of the original fabric, indicating an improvement in the elasticity of the viscose fiber fabric following the fixing treatment.

## 4. Conclusions

This study used N210, HDI, IPDI, MDEA, and TMP to synthesize waterborne polyurethane dye fixative. The comprehensive performance under the conditions of the R-value of 3.0, the MDEA content of 4.0%, the dropwise addition of MDEA, and the TMP content of 1.0% was the best. The application of waterborne polyurethane dye fixative in the fixing treatment of viscose fiber fabrics could apparently improve the color fastness to rubbing. The color fastness to wet rubbing could reach level 3 and to dry rubbing it could reach levels 4-5. The treated fabric demonstrated exceptional durability against color fading. Moreover, the use of waterborne polyurethane dye fixative significantly enhanced the anti-pilling and anti-fuzzing properties, as well as the elasticity, of the viscose fiber fabrics.

## Figures and Tables

**Figure 1 polymers-16-00089-f001:**
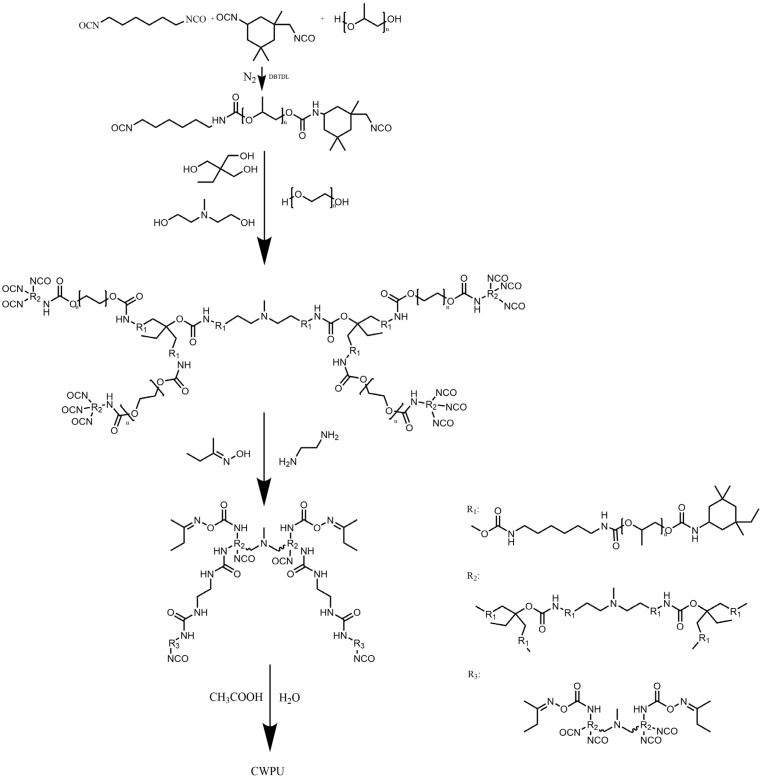
Synthesis steps of CWPU.

**Figure 2 polymers-16-00089-f002:**
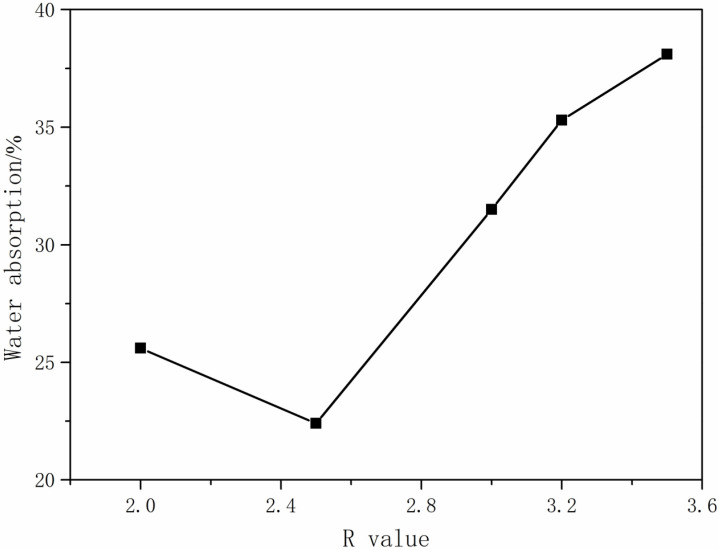
The effect of R-value on the water resistance of adhesive film.

**Figure 3 polymers-16-00089-f003:**
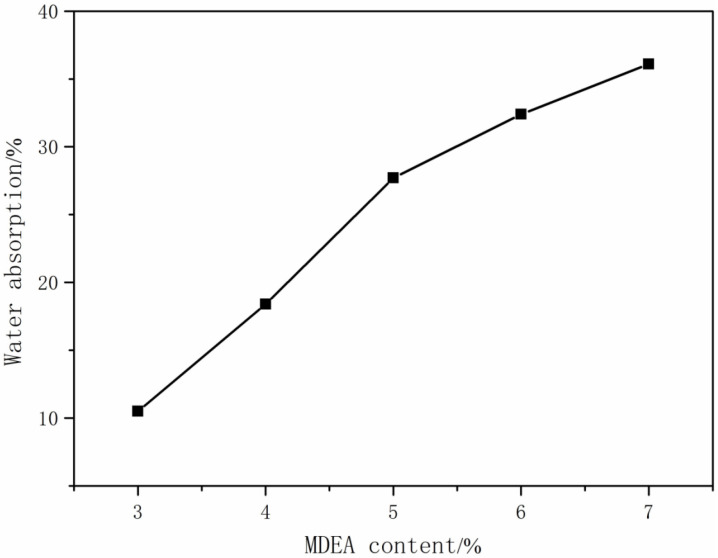
The effect of MDEA content on the water absorption of film.

**Figure 4 polymers-16-00089-f004:**
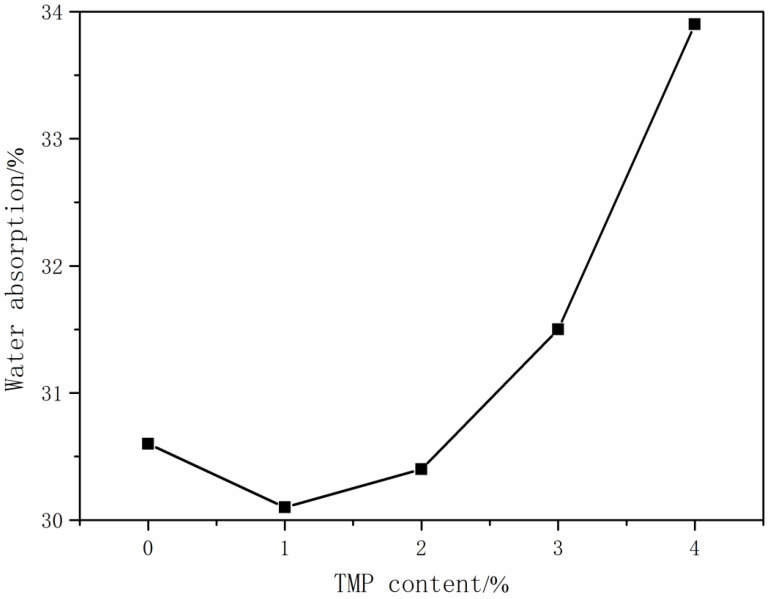
Effect of TMP content on water absorption of membrane.

**Figure 5 polymers-16-00089-f005:**
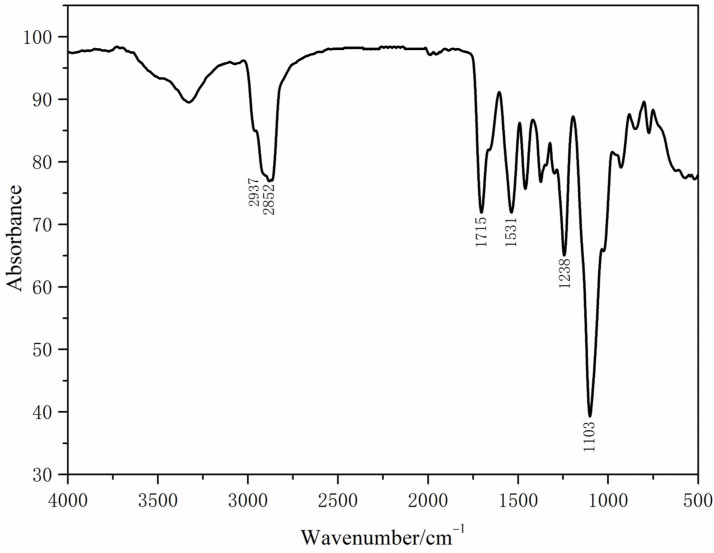
Infrared spectrum of waterborne polyurethane dye fixative film.

**Figure 6 polymers-16-00089-f006:**
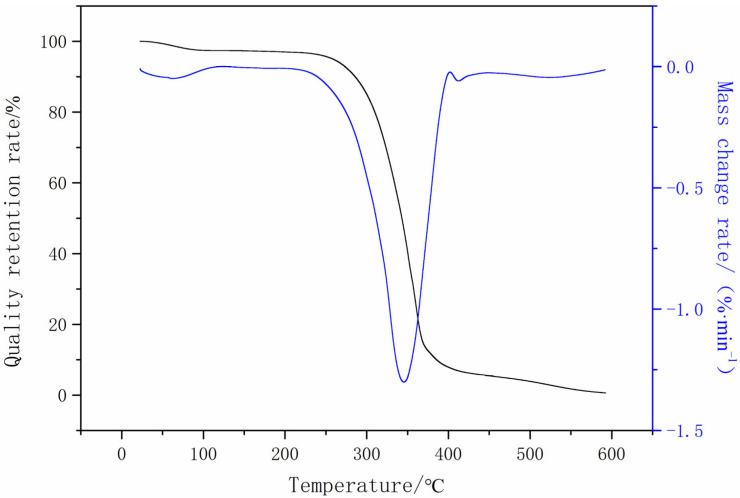
TGA-DTG spectra of cationic waterborne polyurethane.

**Figure 7 polymers-16-00089-f007:**
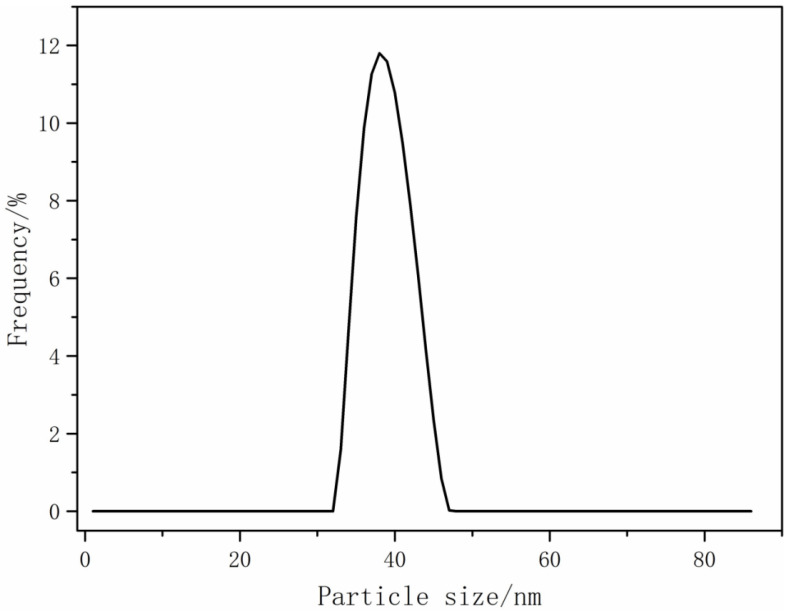
Particle size distribution of waterborne polyurethane dye fixative.

**Table 1 polymers-16-00089-t001:** The influence of R-value on performance of CWPU emulsion.

R-Value	Viscosity (mPa·s)	Stability (60 °C/d)	Appearance of Emulsion	Adhesive Film Feel	Water Absorption Rate (%)
2.0	391	<1	Translucent	Tacky	25.3
2.5	357	<3	Transparent pan-blue light	Tacky	22.4
3.0	307	>7	Transparent pan-blue light	Soft, non-sticky	31.5
3.2	290	>7	Transparent pan-blue light	Soft, non-sticky	35.7
3.5	263	<3	Transparent pan-blue light	Soft, non-sticky	38.4

**Table 2 polymers-16-00089-t002:** Effect of R-value on the viscosity and water absorption properties of CWPU.

R-Value	Pristine Viscosity (mPa·s)	Mean (mPa·s)	Pristine Water Absorption Rate (%)	Mean (%)
2.0	387	391 ± 4.04	25.6	25.3 ± 0.25
	395		25.1	
	390		25.3	
2.5	355	357 ± 6.69	22.4	22.4 ± 0.14
	365		22.4	
	352		22.6	
3.0	306	307 ± 2.64	31.5	31.5 ± 0.30
	310		31.8	
	305		31.2	
3.2	287	290 ± 9.07	35.3	35.7 ± 0.38
	301		35.9	
	284		36.0	
3.5	264	263 ± 2.08	38.1	38.4 ± 0.26
	265		38.5	
	261		38.6	

**Table 3 polymers-16-00089-t003:** Different MDEA addition methods.

Addition Methods	Emulsion State
Directly	The solution is turbid and the viscosity increases, easy to gel
Dropwise addition 10 min	The reaction is stable, many bubbles in the solution
Dropwise addition 20 min	Stable reaction, transparent sol
Dropwise addition 30 min	Stable reaction, transparent sol

**Table 4 polymers-16-00089-t004:** The effect of MDEA content on the performance of CWPU.

MDEA Content %	Viscosity mPa·s	Stability 60 °C/d	Appearance of Emulsion	Adhesive Film Feel	Water Absorption Rate %
3.0	334	<3	Translucent	Soft, non-sticky	10.7
4.0	313	>7	Transparent pan-blue light	Soft, non-sticky	18.2
5.0	323	>7	Transparent pan-blue light	Tacky	27.4
6.0	333	>7	Transparent pan-blue light	Tacky	32.5
7.0	353	>7	Transparent	Tacky	36.5

**Table 5 polymers-16-00089-t005:** Effect of MDEA content on the viscosity and water absorption properties of CWPU.

MDEA Content %	Pristine Viscosity (mPa·s)	Mean (mPa·s)	Pristine Water Absorption Rate (%)	Mean (%)
3.0	331	334 ± 5.38	10.5	10.7 ± 0.44
	341		11.2	
	332		10.6	
4.0	311	313 ± 2.08	18.4	18.2 ± 0.33
	314		18.5	
	315		17.9	
5.0	320	323 ± 2.51	27.7	27.4 ± 0.30
	325		27.4	
	323		27.3	
6.0	328	333 ± 4.58	32.4	32.5 ± 0.26
	337		32.5	
	334		32.7	
7.0	351	353 ± 3.21	36.1	36.5 ± 0.55
	352		37.1	
	357		36.4	

**Table 6 polymers-16-00089-t006:** The effect of TMP content on the performance of CWPU.

TMP Content %	Viscosity mPa·s	Stability 60 °C/d	Appearance of Emulsion	Adhesive Film Feel	Water Absorption Rate %
0.0	290	>7	Transparent pan-blue light	Tacky	30.5
1.0	294	>7	Transparent pan-blue light	Soft, non-sticky	30.3
2.0	306	>7	Transparent pan-blue light	Soft, non-sticky	30.5
3.0	312	>7	Transparent pan-blue light	Soft, non-sticky	31.7
4.0	352	<1	Transparent pan-blue light	Soft, non-sticky	33.9

**Table 7 polymers-16-00089-t007:** Effect of TMP content on the viscosity and water absorption properties of CWPU.

TMP Content %	Pristine Viscosity (mPa·s)	Mean (mPa·s)	Pristine Water Absorption Rate (%)	Mean (%)
0.0	290	290 ± 4.50	30.6	30.5 ± 0.40
	295		30.9	
	286		30.1	
1.0	298	294 ± 3.99	30.1	30.3 ± 0.26
	295		30.6	
	290		30.2	
2.0	304	306 ± 8.07	30.4	30.5 ± 0.17
	301		30.7	
	315		30.6	
3.0	310	312 ± 2.64	31.5	31.7 ± 0.21
	311		31.8	
	315		31.9	
4.0	345	352 ± 6.42	33.9	33.9 ± 0.20
	355		34.1	
	357		33.7	

**Table 8 polymers-16-00089-t008:** The effect of different dye fixative on the finishing effect of viscose fiber fabrics.

Prescription	PUD-1	PUD-2	PUD-3	Original Fabrics
Color fastness to wet rubbing (level)	3	3	3	1–2
Color fastness to dry rubbing (level)	4–5	4–5	4–5	4
Fabric depth value K/S	24.651	24.136	23.641	23.049
∆E	0.63	0.38	0.24	-
∆L*	−0.62	−0.36	−0.15	-
∆a*	−0.11	−0.08	−0.09	-
∆b*	0.02	−0.04	−0.17	-
Resistance to pilling and fuzzing (level)	4	4	4	3–4
Appearance flatness (level)	3	3	3	2–3
Meridian frontal rebound angle (°)	54	48	54	37
Latitudinal frontal rebound angle (°)	114	125	122	141
Total crease recovery angle (°)	168	173	176	141

## Data Availability

Data are contained within the article.

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
