# Peer review of "Synthesis of a Reactive Cationic/Nonionic Waterborne Polyurethane Dye Fixative and Its Application Performance on Viscose Fiber Fabrics"

_polymers, 2023, doi:10.3390/polym16010089_

Round 1

Reviewer 1 Report

Comments and Suggestions for Authors

The research appears to be conventional, and I don't observe any significant novelty in the current study. Regrettably, I cannot recommend the manuscript for publication. Substantial revisions and improvements are required for it to meet the necessary standards. Consequently, I must reject it in its current form.

Comments on the Quality of English Language

The research appears to be conventional, and I don't observe any significant novelty in the current study. Regrettably, I cannot recommend the manuscript for publication. Substantial revisions and improvements are required for it to meet the necessary standards. Consequently, I must reject it in its current form.

Reviewer 2 Report

Comments and Suggestions for Authors

The authors have synthesized a series of cationic waterborne polyurethane (CWPU) emulsions. Further, they have studied the effects of R-value (the ratio of isocyanate groups to hydroxyl groups in pre-polymerization), the content of N-methyldiethanolamine (MDEA) and trimethylolpropane (TMP), and the addition way of MDEA on the properties of CWPU emulsion and film.

Overall, the quality of work looks good. It should be accepted after the following minor revision.

1. Show the pictures of the emulsions formed at different % MDEA and TMP contents.

2. There is a discrepancy in the R-value and water absorption rate (Table 1). At R-value = 2, it shows higher % water absorption (25.6%) than R-value = 2.5 (22.4%), why?

3. The increase in MDEA content (%) increases the water absorption rate (%) (Table 3), what about the effect of TMP content (%) on water absorption rate? Show it in Table 4.

4. The introduction part should be little bit more elaborated highlighting the importance of these type of works and the current state of affairs world-wide.

5. The authors have highlighted what they have performed but they are silent on how their work will be helpful to researchers working in interdisciplinary research areas?

Reviewer 3 Report

Comments and Suggestions for Authors

The authors investigate the optimal conditions for the synthesis of waterborne polyurethane emulsions, which can be used as fixatives on viscose fiber fabrics. The R-value, MDEA content and adding manner, and TMP content on the properties of CWPU emulsion are systematically studied and analyzed. While the results from the designed experiments are discussed properly, there are some comments that I think the authors should address before publication.

1.     Please redraw the Figure 1. The bonding angles are wrong for all the chemical structures.

2.     There is no standard deviation (SD) for all the results presented in the Results and Discussion part. SD should be added for all the results data to confirm the reproducibility of the experiments and results.

3.     There are some details missing in the method section. For example, all the equipment names and models should be added for all the testing.

4.     There are many grammatical errors in the manuscript. 

5. More characterizations for the properties of the resulting CWPU emulsions, such as SEM/TEM, are recommended to observe the emulsion morphology.

Round 2

Reviewer 3 Report

Comments and Suggestions for Authors

I recommend it to be accepted.